# Navigation with Large Language Models:
# Semantic Guesswork as a Heuristic for Planning

**Dhruv Shah**$^{\beta\dagger}$, **Michael Equi**$^{\beta\dagger}$, **Blazej Osinski**$^{\omega}$, **Fei Xia**$^{\gamma}$, **Brian Ichter**$^{\gamma}$, **Sergey Levine**$^{\beta\gamma}$
$^{\beta}$UC Berkeley, $^{\gamma}$Google DeepMind, $^{\omega}$University of Warsaw

**Abstract:** Navigation in unfamiliar environments presents a major challenge for robots: while mapping and planning techniques can be used to build up a representation of the world, quickly discovering a path to a desired goal in unfamiliar settings with such methods often requires lengthy mapping and exploration. Humans can rapidly navigate new environments, particularly indoor environments that are laid out logically, by leveraging semantics — e.g., a kitchen often adjoins a living room, an exit sign indicates the way out, and so forth. Language models can provide robots with such knowledge, but directly using language models to instruct a robot how to reach some destination can also be impractical: while language models might produce a narrative about how to reach some goal, because they are not grounded in real-world observations, this narrative might be arbitrarily wrong. Therefore, in this paper we study how the "semantic guesswork" produced by language models can be utilized as a guiding heuristic for planning algorithms. Our method, Language Frontier Guide (LFG), uses the language model to bias exploration of novel real-world environments by incorporating the semantic knowledge stored in language models as a search heuristic for planning with either topological or metric maps. We evaluate LFG in challenging real-world environments and simulated benchmarks, outperforming uninformed exploration and other ways of using language models.

**Keywords:** navigation, language models, planning, semantic scene understanding

## 1 Introduction

Navigation in complex open-world environments is conventionally viewed as the largely geometric problem of determining collision-free paths that traverse the environment from one location to another. However, real-world environments possess *semantics*. Imagine navigating an airport to get to a terminal: your prior knowledge about the way such buildings are constructed provides extensive guidance, even if this particular airport is unfamiliar to you. Large language models (LLMs) and various language embedding techniques have been studied extensively as ways to interpret the semantics in user-specified *instructions* (e.g., parsing "go to the television in the living room" and grounding it in a specific spatial location), but such models can provide much more assistance in robotic navigation scenarios by capturing rich semantic knowledge about the world. For instance, when looking for a spoon in an unseen house, the LLM can produce a "narrative" explaining why going towards a dishwasher may eventually lead you to find the spoon, and that the robot should prioritize that direction. This is similar to how a person might imagine different ways that the available subgoals might lie on the path to the goal, and start exploring the one for which this "narrative" seems most realistic. However, since language models are not *grounded* in the real world, such models do not know the spatial layout of the robot's surroundings (e.g., there is a couch that the robot needs to circumnavigate). To utilize the semantic knowledge in language models to aid in embodied tasks, we should not just blindly *follow* the language model suggestions, but instead use them as proposals or navigational heuristics. In this paper, we study how that might be accomplished.

We study this idea in the context of visual navigation, where a robot is tasked with reaching a goal denoted by a natural language query $q$ (see Fig. 1) in a *novel* environment using visual observations.

---

$^{\dagger}$ These authors contributed equally. Videos and code: sites.google.com/view/lfg-nav/

7th Conference on Robot Learning (CoRL 2023), Atlanta, USA.

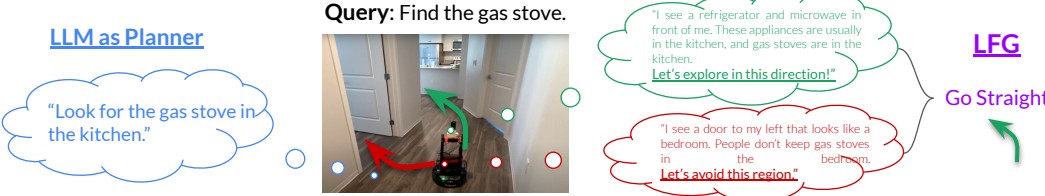

**Figure 1:** In constrast to methods that use LLM plans directly, Language Frontier Guide (LFG) uses a language model to *score* subgoal candidates, and uses these scores to guide a heuristic-based planner.

The robot has no prior experience in the target environment, and must explore the environment to look for the goal. While narratives generated by an LLM may not be sufficient for navigation by themselves, they contain useful cues that can be used to *inform* or *guide* the behavior of the underlying navigation stack for the language navigation task (e.g., by choosing between collision-free subgoal proposals that avoid the couch and lead to the ice tray). We show that this idea can be combined with frontier-based exploration, where the robot maintains a set of unvisited locations at its frontier, *grounds* them in text using a vision-language model (VLM), and *scores* the unvisited subgoals by using LLM reasoning.

We propose Language Frontier Guide, or LFG, a method for leveraging the reasoning capabilities of LLMs to produce a *search heuristic* for guiding exploration of previously unseen real-world environments, combining the strengths of search-based planning with LLM reasoning. LFG is agnostic of the memory representation and planning framework, and can be combined with both (i) a geometric navigation pipeline, building a metric map of the environment for planning and using a hand-designed controller, as well as (ii) a learning-based navigation pipeline, building a topological map for planning and using a learned control policy, yielding a versatile system for navigating to open-vocabulary natural language goals. Our experiments show that LFG can identify and predict simple patterns in previously unseen environments to accelerate goal-directed exploration. We show that LFG outperforms other LLM-based approaches for semantic goal-finding in challenging real-world environments and on the Habitat ObjectNav benchmark.

## 2 Related Work

**Vision-based navigation:** Navigation is conventionally approached as a largely geometric problem, where the aim is to map an environment and use that map to find a path to a goal location [1]. Learning-based approaches can exploit patterns in the training environments, particularly by learning vision-based navigation strategies through reinforcement or imitation [2–7]. Our work is also related to PONI [7], which uses a learned potential function to prioritize frontier points to explore; instead, we use a language model to rank these points. Notably, these methods do not benefit from prior semantic knowledge (e.g., from the web), and must rely entirely on patterns discovered from offline or online navigational data. Our aim is specifically to bring semantic knowledge into navigation, to enable robots to more effectively search for a goal in a new environment.

**Semantic knowledge-guided navigation:** Prior knowledge about the semantics of indoor environments can provide significantly richer guidance. With the advent of effective open-vocabulary vision models [8, 9], some works have recently explored incorporating their semantic knowledge into models for navigation and other robotic tasks with the express aim of improving performance at *instruction following* [10–14]. In general within robotics, such methods have either utilized pre-trained vision-language representations [15–17], or used language models directly to make decisions [18–23]. Our aim is somewhat different: while we also focus on language-specified goals, we are primarily concerned with utilizing the semantics in pre-trained language models to help a robot figure out how to actually reach the goal, rather than utilizing the language models to more effectively interpret a language instruction. While language models can output reasonable substeps for temporally extended tasks in some settings [24, 25], there is contradictory evidence about their ability to actually plan [26], and because they are unaware of the observations and layout in a particular environment, their "plans" depend entirely on the context that is provided to them. In contrast to prior work, our approach does not rely on the language model producing a *good* plan, but merely a heuristic that can bias a dedicated planner to reach a goal more effectively. In this way, we use the language models more to produce *suggestions* rather than actual plans.

**LLM-guided navigation:** Some works have sought to combine predictions from language models with either planning or probabilistic inference [14, 27], so as to not rely entirely on forward prediction from the language model to take actions. However, these methods are more aimed at filtering out *infeasible* decisions, for example by disallowing actions that a robot is incapable of performing, and still focus largely on being able to interpret and process instructions, rather than using the language model as a source of semantic hints. In contrast, by incorporating language model suggestions as heuristics into a heuristic planner, our approach can completely override the language model predictions if they are incorrect, while still making use of them if they point the way to the goal.

Another branch of recent research [28–30] has taken a different approach to ground language models, by making it possible for them to read in image observations directly. While this represents a promising alternative approach to make language models more useful for embodied decision making, we believe it is largely orthogonal and complementary to our work: although vision-language models can produce more grounded inferences about the actions a robot should take, they are still limited only to *guessing* when placed in unfamiliar environments. Therefore, although we use ungrounded language-only models in our evaluation, we expect that our method could be combined with vision-language models easily, and would provide complementary benefits.

## 3   Problem Formulation and Overview

Our objective is to design a high-level planner that takes as input a natural language query $q$ (e.g., "find the bedside table"), explores the environment in search of the queried object, and commands a low-level policy to control a robotic agent. To do this, we maintain an episodic memory of the environment $\mathcal{M}$ in the form of either (i) a 2D map of the environment, where grid cells contain information about occupancy and semantic labels, or (ii) a topological map of the environment, where nodes contain images captured by the robot and corresponding object labels. One way to solve this task is Frontier-Based Exploration (FBE) [31], where a robot maintains a set of unexplored *frontiers* in it's memory, and explores randomly to reach the goal. *Can we do better with access to LLMs?*

We distill the language-guided exploration task to a heuristic-based search problem, where the robot must propose unvisited subgoals or waypoints, score them, and then use a search algorithm (e.g., A*) to plan a path to the goal. Thus, at the core of LFG is the task of *scoring* subgoal proposals. Formally, let's assume we have the task by query $q$, a partially explored environment stored in $\mathcal{M}$, and a set $\mathcal{S}$ of $n$ textual subgoal proposals $s_1, s_2, \ldots, s_n$ (e.g., "a corner with a dishwasher and refrigerator", "a hallway with a door", etc.). Our goal is to score these subgoal proposals with $p(s_i, q, \mathcal{M})$, the probability that the candidate $s_i \in \mathcal{S}$ will lead to the goal $q$ given the current state of the environment, described through $\mathcal{M}$.

We posit that we can leverage the semantic reasoning capabilities of LLMs by prompting them to construct narratives about which semantic regions of the environment are most (and least) likely to lead to the goal. While the narrative itself might be ungrounded, since the LLM knows very little about the environment, reasoning over objects and semantic regions of the environment often generalizes very broadly. For example, even without seeing a new apartment, a human would *guess* that the dining area is close to the kitchen. Hence, rather than directly using LLM scores for planning [23, 25], we incorporate them as a goal-directed *heuristic* to inform the search process. This has two distinct advantages: (i) when the LLM is right, it nudges the search towards the goal, and when it is wrong (or uncertain), we can still default to the underlying FBE algorithm, allowing recovery from LLM failures, and (ii) it allows us to combine the signal from LLMs with other scores that may be more grounded, e.g. distance to subgoals, making the system more versatile.

## 4   LFG: Scoring Subgoals by Polling LLMs

Our aim in this section is to derive a scoring function from LLMs that takes a textual description of subgoal candidates $s_i$ and the goal query $q$ as inputs, and predicts task-relevant probability $p(s_i, q, \mathcal{M})$, conditioned on the episodic memory $\mathcal{M}$. While we may obtain this from next-token likelihoods (or "logprobs"), they do not represent the desired task-relevant probability $p(s_i, q, \mathcal{M})$, and fail to assign similar scores, say, to different subgoals that are semantically similar but have different tokenizations (see our experiments in Section 6 for a comparison). Furthermore, most ca-

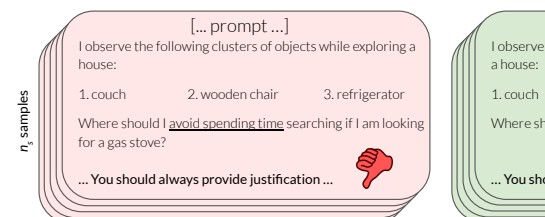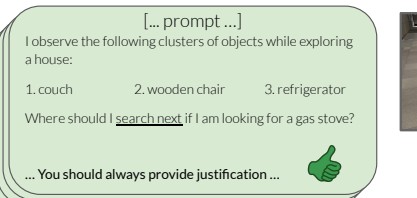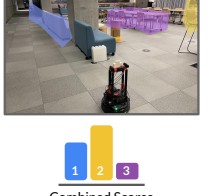

**Figure 2:** LFG scores subgoals with an empirical estimate of the likelihoods by sampling an LLM $n_s$ times with both positive and negative prompts, and uses chain-of-thought to obtain reliable scores. These scores are used by a high-level planner as *heuristics* to guide search. For full prompts, see Appendix B.

pable LLMs of today are available through APIs that do not expose the ability to query logprobs.[1] And lastly, even if reliable logprobs were available, they are incompatible with chain-of-thought prompting [32], which we find to be crucial to success in spatial reasoning.

To overcome these challenges, LFG uses a novel approach to extract task-relevant likelihoods from LLMs. Given candidate subgoal images, LFG uses a VLM to obtain a textual subgoal desriptor $s_i$, which must be scored with the LLM. LFG *polls* the LLMs by sampling the most likely subgoal $n_s$ times, conditioned on a task-relevant prompt. We then use these samples to empirically estimate the likelihood of each subgoal. To get informative and robust likelihood estimates, we use a chain-of-thought prompting (CoT) technique [32], to improve the quality and interpretability of the scores, and use a combination of positive and negative prompts to gather unbiased likelihood estimates. Figure 2 outlines our scoring technique, with the full prompt provided in Appendix B. We now describe the details of our scoring technique.

**Structured query:** We rely on in-context learning by providing an example of a structured query-response pair to the LLM, and ask it to pick the most likely subgoal that satisfies the query. To sample a subgoal from $\mathcal{S}$ using a language model, we prompt it to generate a structured response, ending with ``Answer: i''. This structure allows us to always sample a *valid* subgoal, without having to ground LLM generations in the environment [24].

**Positives and negatives:** We find that only using positive prompts (e.g., "which subgoal is most likely to reach the goal") leads to likelihood estimates being uninformative for cases where the LLM is not confident about any subgoal. To overcome this, we also use negative prompts (e.g., "which subgoal is least likely to be relevant for the goal"), which allows us to score subgoals by eliminating/downweighting subgoals that are clearly irrelevant. We then use the difference between the positive and negative likelihoods to rank subgoals.

**Chain-of-thought prompting:** A crucial component of getting interpretable and reliable likelihood estimates is to encourage the LLM to *justify* its choice by chain-of-thought prompting. As demonstrated in prior works, CoT elicits interpretability and reasoning capabilities in LLMs, and while we don't explicitly use the generated reasonings in our approach (great future work direction), we find that CoT improves the quality and consistency of the likelihood estimates. Additionally, it also helps maintain interpretability, to better understand why the LFG-equipped agent

---

**Algorithm 1:** Scoring Subgoals with LFG

**Data:** Subgoal descriptors $\{l_i \forall s_i \in \mathcal{S}\}$
1   pPrompt ← PositivePrompt($\{l_i\}$)
2   nPrompt ← NegativePrompt($\{l_i\}$)
3   pSamples ← [sampleLLM(pPrompt) $* n_s$]
4   nSamples ← [sampleLLM(nPrompt) $* n_s$]
5   pScores ← sum(pSamples) / $n_s$
6   nScores ← sum(nSamples) / $n_s$
7   **return** pScores, nScores

---

takes certain decisions.

In summary, we score subgoals by sampling the LLM multiple times and empirically estimating the likelihood of each subgoal. We use a combination of positive and negative prompts to get unbiased likelihood estimates, and use chain-of-thought prompting to improve the quality and interpretability of the scores (Figure 2). We will now discuss how these scores can be incorporated into a navigation system as *search heuristics*.

---

[1]Most notably, OpenAI's Chat API for GPT-3.5 and GPT-4, Google's PaLM API, and Anthropic's Claude API all do not support logprobs.

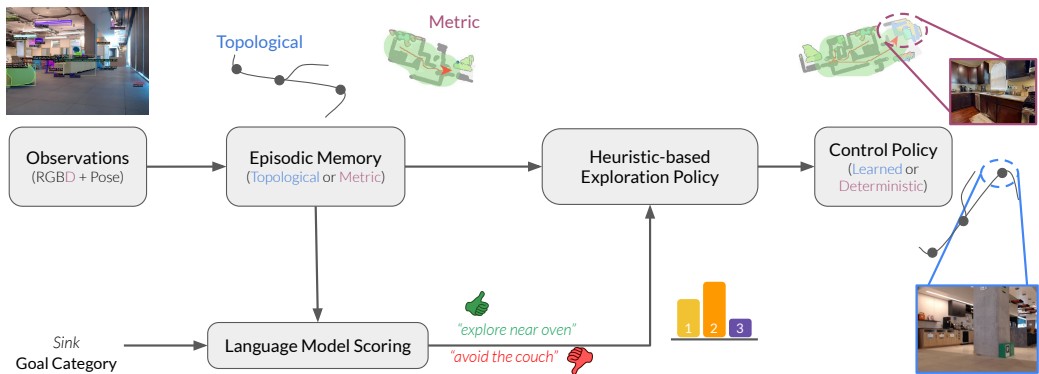

**Figure 3:** Overview of LFG for language-guided exploration. Based on the pose and observations, LFG builds an episodic memory (topological or metric), which is used by the heuristic-based exploration policy to rank adjacent clusters, or subgoal frontiers. Navigation to the subgoal frontier is completed by a low-level policy.

## 5  LLM Heuristics for Goal-Directed Exploration

Given the LLM scoring pipeline outlined in the previous section, our key insight is that we can incorporate these scores in a search-based planning pipeline to *heuristically* guide the search process. We instantiate LFG using frontier-based exploration (FBE) and LLM scores generated via polling.

**FBE:** This method maintains a "map" of the seen parts of the environment, which may be geometric [33] or topological [34], and a frontier separating it from the unexplored parts. By navigating to the *nearest* point of the frontier, the robot explores new areas of the environment until it finds the goal object or completes exploration without finding it. A standard FBE implementation is presented in Algorithm 2 inblack text. The robot maintains either a 2D metric map of its surroundings, or a topological map whose nodes are comprised of the robot's visual observations and edges represent paths taken in the environment. Additionally, we also store semantic labels corresponding to objects detected in the robot's observations, which are used to ground the observations in text.

At a fixed re-planning rate, the high-level planner computes its frontier $f_i$ (Line 10), and picks the frontier point that is *closest* to the current location, i.e., maximizing the distance score (Line 16), and then navigates to this frontier (Line 21). At any point in this process, if the agent's semantic detector detects an object of the same category as the query $q$, it navigates directly to this object and the trajectory ends.

**Incorporating LLM scores:** Our method, LFG, extends FBE by using an additional search heuristic obtained by polling LLMs for semantic "scores". The modifications to FBE are marked in purple in Algorithm 2. After enumerating the frontiers, LFG uses the semantic labels from a VLM [35] to *ground* subgoal images at each frontier $f_i$ (Line 11). These images are converted into textual strings, and form the subgoal candidates $s_i$ that can be scored using Algorithm 1. We associate each frontier point $f_i$ with the nearest object cluster $c_i$ (Line 17), and compute LLM scores for each point as follows:

$$h(f_i, q) = w_p \cdot \text{LLM}_{\text{pos}}(c_i) - w_n \cdot \text{LLM}_{\text{neg}}(c_i) - \text{dist}(f_i, p), \tag{1}$$

where $p$ is the current position of the agent, and $w_p, w_n$ are hyperparameters (see Appendix A.1). We then choose the frontier with the highest score to be the next subgoal (Line 21), navigate to it using a local controller, and repeat the planning process. Algorithm 2 outlines the general recipe for integrating LLM scores as a planning heuristic. Please see Appendix A for specific instantiations of this system with geometric and topological maps, and more details about the referenced subroutines.

## 6  System Evaluation

We now evaluate the performance of LFG for the task of goal-directed exploration in real-world environments, and benchmark its performance against baselines. We instantiate two systems with LFG: a real-world system that uses a topological map and a learned control policy, and a simulated agent that uses a geometric map and a deterministic control policy. Our experiments show that both

**Algorithm 2:** Instantiating LFG for Goal-Directed Exploration

**Data:** $o_0$, Goal language query $q$

1   subgoal ← None
2   **while** *not done* **do**
3      $o_t$ ← getObservation()
4      episodicMemory ← mappingModule($o_t$)
5      **if** *q in semanticMap* **then**
6         subGoal ← getLocation(episodicMemory, q)
7      **else**
8         **if** *numSteps % $\tau$ == 0* **then**
            // replanning
9             location ← getCurrentLocation()
10            frontier ← getFrontier(episodicMemory)
11            objectClusters ← getSemanticLabels(episodicMemory, frontier)
12            $LLM_{pos}, LLM_{neg}$ ← ScoreSubgoals(objectClusters)
13            scores ← []
14            **for** *point in frontier* **do**
15               distance ← DistTo(location, point)
16               scores[point] ← – distance
17               closestCluster ← getClosestCluster(objectClusters, point)
18               $i$ ← clusterID(closestCluster)
19               **if** *dist(closestCluster, point) < $\delta$* **then**
                 // incorporate language scores
20                 scores[point] ← $w_p \cdot LLM_{pos}[i] - w_n \cdot LLM_{neg}[i]$ - distance
21          subgoal ← argmax(scores)
22      numSteps ← numSteps +1
23      goTo(subGoal)

these systems outperform existing LLM-based exploration algorithms by a wide margin, owing to the high quality scores incorporated as search heuristics.

## 6.1   Benchmarking ObjectNav Performance

We benchmark the performance of LFG for the task of object-goal navigation on the Habitat ObjectNav Challenge [36], where the agent is placed into a simulated environment with photo-realistic graphics, and is tasked with finding a query object from one of 10 categories (e.g., "toilet", "bed", "couch" etc.). The simulated agent has access to egocentric RGBD observations and accurate pose information. We run 10 evaluation episodes per scene and report two metrics: the average success rate, and success weighted by optimal path length (SPL), the default metrics for the benchmark. Since LFG requires no training, we do not use the training scenes from HM3D.

We compare to three classes of published baselines: (i) learning-based baselines that learn navigation behavior from demonstrations or online experience in the training scenes [37] on up to 2.5B frames of experience, (ii) search-based baselines [33, 38], and (iii) LLM-based baselines that do not use the training data directly, but leverage the semantic knowledge inside foundation models to guide embodied tasks [18, 39].

Evaluating LFG on the HM3D benchmark, we find that it significantly outperforms search and LLM-based baselines (Table 1). Greedy LLM struggles on the task due to several LLM planning failures, causing the episodes to fail. LFG significantly outperforms the vanilla FBE baseline by leveraging semantic priors from LLMs to score subgoals intelligently. Comparing to learning-based baselines, we find that LFG outperforms most of them and closely matches the state-of-the-art on the task, proving the competence of our polling and heuristic approach. Figure 4 shows an example of the LFG agent successfully reaching the goal by using chain-of-thought and negative prompting.

L3MVN [39], which uses a combination of LLMs and search, performs slightly better than FBE, but is unable to fully leverage the semantics in LLMs. While being similar to LFG, it suffers from

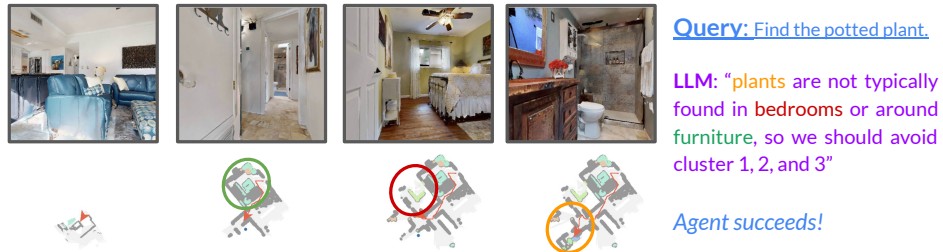

**Figure 4: Qualitative example of a negative score influencing the agent's decision.** LFG discourages the agent from exploring the bedroom and living room, leading to fast convergence toward the goal, whereas FBE fails. The CoT reasoning given by the LLM is shown in purple, justifying its score.

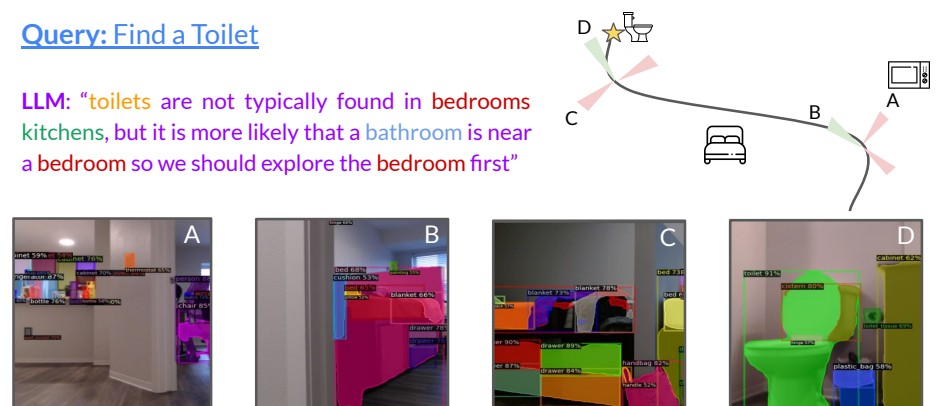

**Figure 5: Qualitative example of LFG in real.** LFG reasons about floor plans in the environment it is searching. In this apartment experiment, LFG believes that a bathroom is more likely to be found near a bedroom rather than a kitchen, and guides the robot towards the bedroom, successfully reaching the goal.

two key limitations: (i) it uses a small language model (GPT-2), which likely does not contain strong semantic priors for the agent to leverage, and (ii) it uses a simple likelihood-based scoring scheme, which we show below is not very effective. Another closely related work, LGX [18], uses a variant of greedy LLM scoring, and hence fails to perform reliably on the benchmark.

Probing deeper into the strong performance of LFG, we ablated various components of our scoring pipeline and studied the change in performance. Note that LGX (Greedy) and L3MVN (No CoT, Logprobs) can be seen as ablations of LFG. Table 2 shows that modifying both the prompting and scoring mechanisms used by LFG lead to large drops in performance. Most notably, scoring via polling ($+7.8\%$) and CoT ($+6.6\%$) are both essential to the strong performance of LFG. Furthermore, we find that using only positive prompts also hurts performance ($-4.7\%$). Popular approaches for using LLMs for planning are significantly outperformed by LFG: Greedy ($-14.5\%$) and Logprobs ($-8.5\%$). Figure 4 shows an example of the LFG agent successfully reaching the goal by using CoT and negative prompting.

**Setup:** For these experiments, we mimic the semantic mapping pipeline of the best-performing baseline on the benchmark [33, 38], and integrate LFG with the geometric map. The simulated agent builds a 2D semantic map of its environment, where grid cells represent both occupancy and semantic labels corresponding to objects detected by the agent. Prior work has shown that state-of-the-art vision models, such as DETIC, work poorly in simulation due to rendering artifacts [33]; hence, we use ground-truth semantic information for all simulated baselines to analyze navigation performance under perfect perception.

## 6.2 Real-world Exploration with LFG

To show the versatility of the LFG scoring framework, we further integrated it with a heuristic-based exploration framework that uses topological graphs for episodic memory [34]. We compare two published baselines: a language-agnostic FBE baseline [40], and an LLM-based baseline that uses the language model to greedily pick the frontier [18].

| Method | Success | SPL | Data |
|---|---|---|---|
| DD-PPO [37] | 27.9 | 14.2 | 2.5B |
| FBE [33] | 61.1 | 34.0 | 0 |
| SemExp [38] | 63.1 | 0.29 | 10M |
| OVRL-v2 [42] | 64.7 | 28.1 | 12M |
| Greedy LLM [18] | 54.4 | 26.9 | **0** |
| L3MVN [39] | 62.4 | | **0** |
| LFG (Ours) | **68.9** | **36.0** | **0** |

| Method | Success | Δ |
|---|---|---|
| LFG (Full) | **68.9** | – |
| **Prompting** | | |
| No CoT | 62.3 | (6.6) |
| Only Positives | 64.2 | (4.7) |
| **Scoring** | | |
| Random | 61.1 | (7.8) |
| Logprobs | 60.4 | (8.5) |
| Ask | 62.4 | (6.5) |

**Table 1:** LFG outperforms all LLM-based baselines on HM3D ObjectNav benchmark, and can achieve close to SOTA performance without any pre-training.

**Table 2:** We find that CoT prompting with positives and negatives, combined with polling, are essential to achieve the best performance.

We evaluate this system in two challenging real-world environments: a cluttered cafeteria and an apartment building (shown in Figures 3 and 5). In each environment, the robot is tasked to reach an object described by a textual string (e.g., "kitchen sink" or "oven"), and we measure the success rate and efficiency of reaching the goal. Episodes that take longer than 30 minutes are marked as failure. While we only tested our system with goal strings corresponding to the 20,000 classes supported by our object detector [35], this can be extended to more flexible goal specifications with the rapid progress in vision-language models.

We find that the complexity of real-world environments causes the language-agnostic FBE baseline to *time out*, i.e., the robot is unable to reach the goal by randomly exploring the environment. Both LLM baselines are able to leverage the stored semantic knowledge to guide the exploration in novel environments, but LFG achieves 16% better performance. Figure 5 shows an example rollout in a real apartment, where the robot uses LFG to reach the toilet successfully.

**Setup:** We instantiate LFG in the real-world using a previously published topological navigation framework [34] that builds a topological map of the environment, where nodes correspond to the robot's visual observations and edges correspond to paths traversed in the environment. This system relies on omnidirectional RGB observations and does not attempt to make a dense geometric map of the environment. To obtain "semantic frontiers" from the omnidirectional camera, we generate $n_v = 4$ *views* and run an off-the-shelf object detector [35] to generate rich semantic labels describing objects in these views. The robot maintains a topological graph of these views and semantic labels, and picks the frontier view with the highest score (Algorithm 2, Line 21) according to LFG. The robot then uses a Transformer-based policy [6, 41] to reach this subgoal. For more implementation details, see Appendix A.3.

# 7  Discussion

We presented LFG, a method for utilizing language models for *semantic guesswork* to help navigate to goals in new and unfamiliar environments. The central idea in our work is that, while language models can bring to bear rich semantic understanding, their ungrounded inferences about how to perform navigational tasks are better used as suggestions and heuristics rather than plans. We formulate a way to derive a heuristic score from language models that we can then incorporate into a planning algorithm, and use this heuristic planner to reach goals in new environments more effectively. This way of using language models benefits from their inferences when they are correct, and reverts to a more conventional unguided search when they are not.

**Limitations and future work:** While our experiments provide a validation of our key hypothesis, they have a number of limitations. First, we only test in indoor environments in both sim and real yet the role of semantics in navigation likely differs drastically across domains – e.g., navigating a forest might implicate semantics very differently than navigating an apartment building. Exploring the applicability of semantics derived from language models in other settings would be another promising and exciting direction for future work. Second, we acknowledge that multiple requests to cloud-hosted LLMs with CoT is slow and requires an internet connection, severely limiting the extent of real-world deployment of the proposed method. We hope that ongoing advancements in quantizing LLMs for edge deployment and fast inference will address this limitation.

**Acknowledgments**

This research was partly supported by AFOSR FA9550-22-1-0273 and DARPA ANSR. The authors would like to thank Bangguo Yu, Vishnu Sashank Dorbala, Mukul Khanna, Theophile Gervet, and Chris Paxton, for their help in reproducing baselines. The authors would also like to thank Ajay Sridhar, for supporting real-world experiments, and Devendra Singh Chaplot, Jie Tan, Peng Xu, and Tingnan Zhang, for useful discussions in various stages of the project.

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

# A Implementation Details

## A.1 Hyperparameters

| | Parameter | Value |
|---|---|---|
| $\tau$ | Replanning Rate | 1 |
| $\delta$ | Language Influence Threshold | 2m |
| $n_s$ | Number of LLM Samples | 10 |
| $w_p$ | Weight of Positive Scores | 300 |
| $w_n$ | Weight of Negative Scores | 150 |
| | Max Time Steps | 500 |

**Table 3:** Hyperparameters

## A.2 Computational Resources

| Parameter | Value |
|---|---|
| LLM | gpt-3.5-turbo[2] |
| Evaluation Runtime | 5 hours |
| Compute Resources | $4 \times V100$ |
| Total LLM Tokens | 30M |
| Average API Cost | 15 USD |

**Table 4:** Parameters and resources required to run one evaluation round of LFG on the benchmark.

## A.3 Real World Results

**Generating Prompts:** For both topological and geometric maps we use hand engineered methods for clustering objects in ways that the LLM can efficiently reason over. For geometric maps we implement two functions: $parseObjects$ and $clusterObjects$. In our implementation, $parseObejcts$ filters the geometric map and identifies the cluster centers among each class. $clusterObjects$ takes the cluster centers and performs agglomerative clustering with a threshold of 6 meters, which is roughly the size of one section of a standard house. For topological maps we rely on the configuration of the four cameras to automatically perform parsing and clustering. In our implementation all the objects detected in each frame from either the front, left, right, or rear facing cameras is considered a single cluster.

**Perception:** For the hardware, we use a locobot base with a four HD logitech web cameras that are positioned at 90 degrees relative to each other. At each step of LFG each of four cameras is recorded and frames are semantically annotated. LFG directly uses these frames to determine if the robot should continue to move forward, turn left, turn right, or turn around a full 180 degrees. To improve the performance of our system we choose to whitelist a subset of the 20,000 classes. This reduces the size of the API calls to the language models and helps steer the LLM to focus on more useful information. Following is the complete whitelist used in our experiments:

- toaster
- projector
- chair
- kitchen table
- sink
- kitchen sink
- water faucet
- faucet
- microwave oven
- toaster oven
- oven
- coffee table
- coffee maker
- coffeepot
- dining table
- table
- bathtub
- bath towel

- urinal
- toilet
- toilet tissue
- refrigerator
- automatic washer
- washbasin
- dishwasher
- television set
- sofa
- sofa bed
- bed

- chandelier
- ottoman
- dresser
- curtain
- shower curtain
- trash can
- garbage
- cabinet
- file cabinet
- monitor (computer equipment)

- computer monitor
- computer keyboard
- laptop computer
- desk
- stool
- hand towel
- shampoo
- soap
- drawer
- pillow

**Low-level Policy:** The low-level policy running on the robot is the NoMaD goal-conditioned diffusion policy trained to avoid obstacles during exploration and determine which frontiers can be explored further [41].

**High-level Planning:** For real-world experiments, we follow the setup of ViKiNG [34], where the agent runs a simple frontier-based exploration algorithm and incorporates the LLM scores as *goal-directed heuristics* to pick the best subgoal frontier. For simulation experiments, we use a geometric map coupled with frontier-based exploration, following the setup of Chaplot et al. [38]. Algorithms 3 and 4 summarize the high-level planning module in both cases.

---

**Algorithm 3:** Instantiating LFG with Topological Mapping

**Data:** $o_0$, Goal language query $q$
1  subgoal ← None
2  **while** *not done* **do**
3     $o_t$ ← getObservation()
4     frontierPoints ← mappingModule($o_t$)
5     **if** *q in frontierPoints* **then**
6        turnTowardGoal(frontierPoints)
7     **else**
8        **if** *numSteps* $\% \ \tau == 0$ **then**
9           location ← getCurrentLocation()
10          $LLM_{pos}, LLM_{neg}$ ← scoreFrontiers(frontierPoints) scores ← []
11          **for** *point in frontier* **do**
12             distance ← distTo(location, point)
13             scores[point] ← $w_p \cdot LLM_{pos}\,[i]$ - $w_n \cdot LLM_{neg}\,[i]$ - distance
14          subgoal ← argmax(scores)
15          numSteps ← numSteps +1
16          goTo(subGoal)

---

## A.4 More Experiment Rollouts

Figure 6 shows an example where the negative scoring is essential to LFG's success. Figures 7 and 8 show examples of LFG deployed in a previously unseen apartment and an office building, successfully exploring the environments to find an oven and a kitchen sink.

# B  Prompts

## B.1  Positive Prompt

**Algorithm 4:** Instantiating LFG with Geometric Mapping

**Data:** $o_0$, Goal language query $q$

```
1  subgoal ← None
2  while not done do
3  │   o_t ← getObservation()
4  │   obstacleMap, semanticMap ← mappingModule(o_t[depth], o_t[semantic])
5  │   if q in semanticMap then
6  │   │   subGoal ← getLocation(semanticMap, q)
7  │   else
8  │   │   if numSteps % τ == 0 then
   │   │   │   // replanning
9  │   │   │   location ← getCurrentLocation()
10 │   │   │   frontier ← getFrontier(obstacleMap)
11 │   │   │   objects ← parseObjects(semanticMap)
12 │   │   │   objectClusters ← clusterObjects(objects)
13 │   │   │   LLM_pos, LLM_neg ← ScoreSubgoals(objectClusters)
14 │   │   │   scores ← []
15 │   │   │   for point in frontier do
16 │   │   │   │   distance ← distTo(location, point)
17 │   │   │   │   scores[point] ← – distance
18 │   │   │   │   closestCluster ← getClosestCluster(objectClusters, point)
19 │   │   │   │   i ← clusterID(closestCluster)
20 │   │   │   │   if dist(closestCluster, point) < δ then
   │   │   │   │   │   // incorporate language scores
21 │   │   │   │   │   scores[point] ← w_p· LLM_pos[i] - w_n · LLM_neg[i] - distance
22 │   │   │   subgoal ← argmax(scores)
23 │   │   numSteps ← numSteps +1
24 │   │   goTo(subGoal)
```

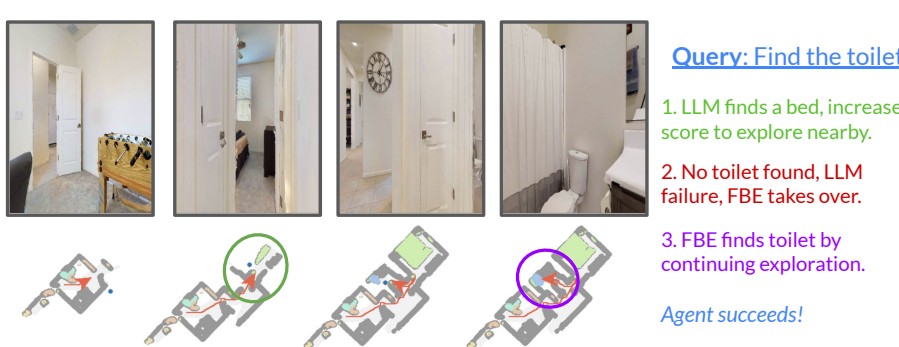

**Query:** Find the toilet.

1. LLM finds a bed, increases score to explore nearby.

2. No toilet found, LLM failure, FBE takes over.

3. FBE finds toilet by continuing exploration.

*Agent succeeds!*

**Figure 6: Tolerance to LLM failures.** An example rollout of LFG compensating for LLM failure. FBE takes over in this case and eventually succeeds, whereas the Greedy agent fails.

```
You are a robot exploring an environment for the first time.
   You will be given an object to look for and should provide
   guidance of where to explore based on a series of
   observations. Observations will be given as a list of
   object clusters numbered 1 to N.

Your job is to provide guidance about where we should explore
   next. For example if we are in a house and looking for a tv
    we should explore areas that typically have tv's such as
   bedrooms and living rooms.
```

**Query:** Find an Oven

**LLM**: "Ovens are typically found in kitchens. Cabinets and microwaves are typically found in kitchens so we should explore toward those objects."

**Figure 7: LFG in an unseen apartment.** The robot starts in the same starting location and environment as 5, and is tasked with finding an oven. LFG guides the robot towards the kitchen appliances, rather than the bedroom door, and successfully leads to the oven.

**Query:** Find an Sink

**LLM**: ""A sink will likely be found in a bathroom or a kitchen. Therefore toilet tissue and refrigerators may be near a sink. Let's explore these regions first and if I do not get closer to the goal, I can continue exploring until there is a stronger signal""

Erroneous Detections

Toilet Tissue    Refrigerator

**Figure 8: LFG in an unseen office building.** The agent looks for a sink in an open-plan office building. Despite erroneous detections, the robot continues exploring the environment, with LFG guiding it towards frontiers containing appliances found in a cafe. The robot successfully finds the sink despite imperfect detections.

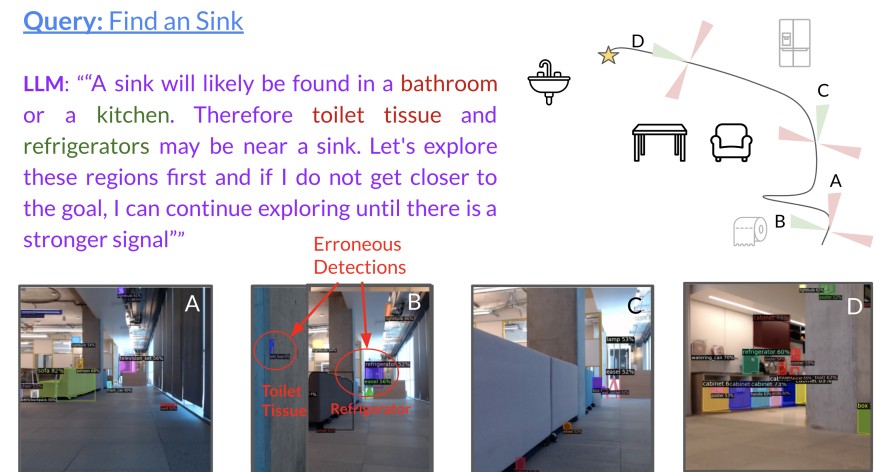

```
You should always provide reasoning along with a number
    identifying where we should explore. If there are multiple
    right answers you should separate them with commas. Always
    include Reasoning: <your reasoning> and Answer: <your
    answer(s)>. If there are no suitable answers leave the
    space afters Answer: blank.

Example

User:
I observe the following clusters of objects while exploring a
    house:

1. sofa, tv, speaker
2. desk, chair, computer
3. sink, microwave, refrigerator
```

Where should I search next if I am looking for a knife?

Assistant:
Reasoning: Knifes are typically kept in the kitchen and a sink,
    microwave, and refrigerator are commonly found in kitchens
    . Therefore we should check the cluster that is likely to
    be a kitchen first.
Answer: 3

Other considerations

1. Disregard the frequency of the objects listed on each line.
   If there are multiple of the same item in  a cluster it
   will only be listed once in that cluster.
2. You will only be given a list of common items found in the
   environment. You will not be given room labels. Use your
   best judgement when determining what room a cluster of
   objects is likely to belong to.

## B.2 Negative Prompt

```
You are a robot exploring an environment for the first time.
    You will be given an object to look for and should provide
    guidance of where to explore based on a series of
    observations. Observations will be given as a list of
    object clusters numbered 1 to N.

Your job is to provide guidance about where we should not waste
    time exploring. For example if we are in a house and
    looking for a tv we should not waste time looking in the
    bathroom. It is your job to point this out.

You should always provide reasoning along with a number
    identifying where we should not explore. If there are
    multiple right answers you should separate them with commas
    . Always include Reasoning: <your reasoning> and Answer: <
    your answer(s)>. If there are no suitable answers leave the
     space afters Answer: blank.

Example

User:
I observe the following clusters of objects while exploring a
    house:

1. sofa, tv, speaker
2. desk, chair, computer
3. sink, microwave, refrigerator

Where should I avoid spending time searching if I am looking
    for a knife?

Assistant:
Reasoning: Knifes are typically not kept in a living room or
    office space which is what the objects in 1 and 2 suggest.
    Therefore you should avoid looking in 1 and 2.
Answer: 1,2

Other considerations

1. Disregard the frequency of the objects listed on each line.
    If there are multiple of the same item in  a cluster it
    will only be listed once in that cluster.
2. You will only be given a list of common items found in the
    environment. You will not be given room labels. Use your
    best judgement when determining what room a cluster of
    objects is likely to belong to.
```

