# OpenReview forum: "Navigation with Large Language Models: Semantic Guesswork as a Heuristic for Planning"
_robot-learning.org/CoRL/2023/Conference — CoRL 2023 Poster_

### Official Review · Reviewer_JLdp · 2023-07-03

**Confidence:** 4
**Originality:** Good
**Technical Quality:** Very Good
**Clarity Of Presentation:** Excellent
**Impact:** 3

**Recommendation:**

Strong Accept: I recommend accepting the paper and will argue for my recommendation even if other reviewers hold a different opinion.

**Review:**

## Strengths

- *Main ideas are interesting and well-motivated*. The core ideas of (1) using an LLM to guide a search-based planner instead of directly planning, and (2) scoring candidates by polling instead of using logprobs are very well motivated, sensible, and interesting. Given the large amount of related work that attempts to use LLM’s to replace planners, I believe (1) is a particularly useful contribution.
- *Clear writing and presentation*. The presentation makes the method simple and quite easy to understand.
- *Solid empirical results* The presented empirical results on the ObjectNav challenge are impressive and rather convincing.

## Weaknesses

- *Lack of a few important, relevant details on the method*. While the authors provide a good high-level picture of their proposed method, there are a number of details that aren’t mentioned in the main paper or the supplementary material. Specifically, I have the following questions about the method:
    - How do the `parseObjects()` and `clusterObjects()` methods in Algorithm 2 work?: These seem like rather important methods, yet they are mentioned only in a cursory fashion.
        - Does `parseObjects()` simply return the ‘semantic label’ associated with every single object in the map? Or is it only the set of objects within the current frontier?
        - Does `clusterObjects()` perform clustering by the objects’ semantic label(s), distance, or both?
            - What sort of clustering algorithm is used? Is the size of each cluster pre-determined?
            - Is there an implicit assumption here that there aren’t that many different objects with different classes in the map/environment? It occurs to me that if there are (which happens, for instance, in the [BEHAVIOR dataset](https://behavior.stanford.edu/)), then the clustering approach might struggle. Moreover, there might be too many objects to even fit into the prompt, given the context-window of the LLM. What exactly happens in the case where there are a large number of objects with potentially very different semantic labels?
    - What exactly is one ‘step’ in Algorithm 2? You mention that replanning is triggered when `numSteps % \tau == 0` , is one step one command to the robot’s motors (occurring at something like 30 Hz), or is it one completed navigation to a particular point? I’d assume it’s the latter because the supplement mentions that tau is set to 1…
    - Is the ‘semantic label’ of an object simply the object category? Does it include any other information?
    - Can the LLM respond with multiple positive and negative subgoals, or only one each? Lines 146 to 150 in the paper make it seem like there can only be one positive and one negative subgoal id output by the LLM, but Section C.2 in the supplementary material show the case where two negative goals are output.
    - What exactly is the `Ask` ablation in Table 2? I couldn’t find a description of it anywhere in the text.
- *Incremental method (minor).* While the presented method is interesting and well-motivated, the application to object-finding feels rather limited. The presented improvements are statistically-significant, but not that much more than existing baselines. It is unclear how this approach could be built upon to enable even more progress on the ObjectNav challenge and the wider object search problem. It would be much more significant and impressive if the authors expanded their technique to navigation and manipulation tasks, perhaps leveraging the LLM for both object-search and for exploring properties towards manipulation of objects (e.g.: leveraging the LLM so that the robot can understand that it is difficult to pick up a chair, but a chair can be dragged, etc.). I acknowledge that this is a much harder problem within a broader research agenda, but it feels like the common-sense knowledge in LLM’s can be utilized for much more than efficient object search. In my opinion, showcasing some aspect of this wider applicability would make the paper’s contribution(s) much more significant .

## Suggestions

In addition to answering all the listed questions and updating the text accordingly, I have the following relatively minor suggestions:

- Consider mentioning approaches that perform navigation and object search by solving POMDP’s (e.g. [this semi-recent work](https://arxiv.org/abs/2005.02878)). Some of these approaches (e.g. [this one](https://ieeexplore.ieee.org/document/8793888)) are even guided by language input, and they seem quite closely-related to your setting.
- Subsection 6.1 appears blank; remove it

**Quality Of The Limitations Section:**

Limitations are addressed clearly

**Questions For Rebuttal:**

In addition to the questions listed under the first bullet point in the above ‘weaknesses’ sub section, I would like to ask:

- Did the authors run an ablation where the number of LLM samples (n_s) is set to 1?
- What is $\phi$ supposed to represent in Algorithm 1?
- Have I misunderstood anything/missed any important details? If so, please point them out; I’m more than happy to adjust my review accordingly!

**Robotics Focus:**

Sufficient demonstration on hardware

**Summary Of Paper:**

This paper presents a method called LFG that uses LLM’s to enable robots to efficiently find objects in previously-unseen environments. The main idea is to leverage the common-sense knowledge from LLM’s as a prior to guide a search-based planning algorithm. This enables the method to efficiently find objects when the LLM outputs useful knowledge while also being able to fall back onto the classical frontier exploration planning algorithm when the LLM outputs are useless or nonsensical. Instead of attempting to use the logprobs of the LLM as many related approaches do, the authors poll the LLM using a specifically-designed prompt and leverage chain-of-thought prompting to improve the quality of polling output. Experiments on tasks from the Habitat ObjectNav Challenge demonstrate the the proposed method is able to perform on par with or better than baselines despite not requiring any training.

**Summary Of Recommendation:**

Overall, this is an interesting and well-written paper that contributes a well-motivated method to the robotics community. Though I found the work slightly incremental, I believe that this is above the bar for acceptance at CoRL provided the paper is updated to provide details answering the various questions I listed above.

---

### Official Review · Reviewer_FZa9 · 2023-07-19

**Confidence:** 5
**Originality:** Good
**Technical Quality:** Very Good
**Clarity Of Presentation:** Good
**Impact:** 3

**Recommendation:**

Weak Accept: I recommend accepting the paper, but will not argue for my recommendation if the majority of other reviewers have a different opinion.

**Review:**

The paper is generally well-written and the approach is sound. The results are also promising and show good performance on a challenging task without any domain-specific training.

The central issue is not with the method or results, but with the language used to describe the method. Multiple places use language that may imply that planning sequential decision making (i.e., via the Bellman Equation) is being used, especially since the non-learned baseline is referred to as a "Greedy" strategy. It seems (see Alg. 2) that the presented LFG approach is *also* a greedy approach, and so naming it such is misleading. [Note: I recognize that "search" is an overloaded term here, but I think it is possible to rework some of the language to make it more easily to understand where the work is situated with respect to other approaches.]

A related suggestion: it might be better if the text (see Line 52, 175) did not use the phrase "search heuristic" to describe the LLM contribution of the LFG approach as algorithmic search is not used, even though the robot is 'searching' the environment. Similarly, it might be more appropriate to characterize the approach as a method of "frontier selection" rather than "planning", as used in lines 200 and 201 (among others). Finally, changing the name of the baseline approach (currently the Greedy approach) to something more appropriate: e.g., non-learned baseline, "nearest-FBE", or similar.

I welcome feedback from the authors on these points, but I think it would be best to rework some of the language to make clearer the role that the LLM guidance plays and what type of search process is used. This would improve the paper and readability.

Other comments and questions:
- Clarification: how were the weights chosen? Why is it that the values differ for the positive and negative weights? How sensitive to those parameters do the authors expect the behavior to be?
- Clarification: the text and appendix are clear about how prompting is done for the purposes of scoring via Alg. 1, but can the authors also clarify the process for computing the "logprobs" (e.g., what prompt was used and what completions provided)?
- There appear to be footnotes 3 and 4 in the Appendix, but they do not seem to correspond to any text.
- Table 2: Please use a symbol other than \pi to reference the in-caption footnote, as \pi is used in planning literature to represent a policy. Instead, a \dagger might be more appropriate and would avoid confusion (e.g., that an asterix * might imply).


**Quality Of The Limitations Section:**

Limitations are addressed clearly

**Questions For Rebuttal:**

[These questions have been reproduced here from my full review above for clarity.]

- Clarification: how were the weights chosen? Why is it that the values differ for the positive and negative weights? How sensitive to those parameters do the authors expect the behavior to be?
- Clarification: the text and appendix are clear about how prompting is done for the purposes of scoring via Alg. 1, but can the authors also clarify the process for computing the "logprobs" (e.g., what prompt was used and what completions provided)?


**Robotics Focus:**

Highly relevant to robotics but no hardware experiments

**Summary Of Paper:**

This paper proposes "Language Frontier Guide" (LFG), an approach to using pretrained large language models (LLM) to aid in navigating through an unknown environment to reach a semantic goal. The method involves using a unique querying method to aggregate responses to both "positive" and "negative" language prompts into scores used to bias frontier-based exploration towards frontiers more likely to lead to the object of interest. Results in the Habitat simulator demonstrate improved performance over both non-learned "FBE" (frontier-based explroration) approaches as well as learned approaches trained specifically for the Habitat environments, demonstrating the potential long-term utility of using off-the-shelf LLMs to inform search behaviors.

**Summary Of Recommendation:**

Overall this is a fairly solid paper with a useful contribution that pushes forwards our understanding of how LLMs will shape robot decision-making. The paper and its results are good, yet the presentation of the material is potentially confusing to a subset of readers, and the authors should take some time to make the language more specific so as to avoid this confusion. A few more clarifications about methodological details should also be included (even if only in the Appendix).

---

### Official Review · Reviewer_4mQY · 2023-07-19

**Confidence:** 4
**Originality:** Good
**Technical Quality:** Very Good
**Clarity Of Presentation:** Excellent
**Impact:** 1

**Recommendation:**

Weak Reject: I recommend rejecting the paper, but will not argue for my recommendation if the majority of other reviewers have a different opinion.

**Review:**

Post-rebuttal:

Thank you for your thorough rebuttal. I appreciate the extensive work the authors have put into their real-world study, and believe this strengthens the claims of the paper.

I am convinced that LFG is an effective heuristic for guiding a robot's navigation using semantic information. The original evaluation was extensive and compelling, and the newly-added experiment in a real-world setting strengthens this claim even further.

I am still not convinced, however, that there is a learning contribution here. While it's true that CoRL has accepted papers that integrate LLMs, those papers proposed new technical contributions via the additional learning algorithms that enable the use/adaptation of LLMs. SayCan, for example, describes an RL algorithm for learning task models that are conditioned on language descriptions.

In contrast, LFG's technical contribution is a planning heuristic. While it is demonstrated to be a highly effective heuristic that could have impact in the planning community, I don't see it having impact as a robot learning paper. However, I am happy to defer to the area chair's view on this (and thus am changing my recommendation to "weak reject").

----------

Strengths:

+ Overall, the paper is very well-organized and easy to read.

+ The evaluation is thorough: it uses an extensive benchmark of simulated navigation tasks and contains comparisons to 8 different baseline methods.

+ The discussion of *how* to formulate LLM prompts (and the comparison between their effectiveness) is interesting and may be useful to other researchers who are interested in integrating LLMs into their work.

Weaknesses:

- My biggest concern is whether this paper is a good fit for CoRL: it does not actually contain any robot learning (but rather, uses a pre-trained LLM).

- The results imply that, compared to learning-based approaches, LFG enables better performance while also requiring zero training data. This is an unfair comparison, however; LFG relies on ground-truth semantics to identify the objects located near the robot, while the learning-based methods do not have this information and presumably must learn it from visual input. I expect that the large amounts of training data are actually needed to enable *visual* learning for these methods, rather than being necessary for the navigation task. Essentially, LFG and the learning-based methods are essentially solving different robotics problems, yet are compared equally in the evaluation.

**Quality Of The Limitations Section:**

Limitations are addressed clearly

**Questions For Rebuttal:**

Clarifications:

* Is there any data showing a more fair comparison to learning-based methods? I.e., where those methods have access to ground-truth semantic information?

* Does the sequence order of explored spaces in M inform the heuristic?

* Line 236: LFG "maxes out the benchmark" -- What does this mean? It doesn't reach full success at the task.

* SPL reflects the success relative to the explored path length. It would be useful to have data about the path length itself in order to assess whether LFG enables more efficient exploration than other methods.

* Section 6.1 is empty. Is 6.2 meant to be a subsection of it? (i.e., 6.2 should be 6.1.1?)

**Robotics Focus:**

Sufficient demonstration on hardware

**Summary Of Paper:**

This paper addresses the problem of robot navigation in unmapped settings. The contribution is the Language Frontier Guide (LFG), which uses LLMs to bias how a robot explores its environment. It polls an LLM to identify the objects in its search frontier that are most likely to be semantically indicative of the search goal. The experimental results show how LFG enables more successful navigation on the Habitat ObjectNav benchmark compared to learning-based and other LLM-based methods.

**Summary Of Recommendation:**

Overall, the paper is well-written and contributes an effective method for navigation in unfamiliar environments. However, it does not contribute any methods for robot learning and thus may not be suited for CoRL. Additionally, the results rely on an unfair comparison in which the proposed method is provided more information than the learning-based baselines.

---

### Official Review · Reviewer_oCzb · 2023-07-19

**Confidence:** 4
**Originality:** Very Good
**Technical Quality:** Very Good
**Clarity Of Presentation:** Very Good
**Impact:** 4

**Recommendation:**

Weak Accept: I recommend accepting the paper, but will not argue for my recommendation if the majority of other reviewers have a different opinion.

**Review:**

Globally, the paper is clear and exploits an original idea. This can pave the way to new methodologies benefiting from LLMs without the burden of grounding language. Furthermore, the results presented  on the HM3D ObjectNav benchmark as well as the ablation study support the contributions of this paper. Minor comments should be addressed by the authors.

**Quality Of The Limitations Section:**

Limitations are addressed clearly

**Questions For Rebuttal:**

1) Please clarify the role of $n_s$ in Algorithm 1. Line 140: It is not clear if each subgoal is sampled $n_s$ times or if $n_s$ is the number of subgoals.
2) Please detail more Figure 2. What is the meaning of 1, 2 and 3?
3) Line 149, please explain "Answer i". What does "i" refer to?
4) There may be situations where all subgoals are not informative to reach the destination (i.e. low score). How is this handled by the proposed algorithm?

**Robotics Focus:**

Highly relevant to robotics but no hardware experiments

**Summary Of Paper:**

This paper proposes LFG, a heuristic planning algorithm exploiting semantic reasoning from LLM. The method more exactly extends Frontier-based exploration algorithm with semantic knowledge to navigate in unknown environments. The authors exploit prompts from LLM to score positively or negatively subgoals fed to an exploration policy. As so, the approach does not require any training and is validated on the HM3D ObjectNav Benchmark.


**Summary Of Recommendation:**

The paper is clear, original and interesting for the community. The reviewers recommends the paper for CoRL.

---

### Author Response · Authors · 2023-08-15
**Summary of Discussion Phase**

All reviewers acknowledged originality of contributions (oCzb, JLdp), technical quality, organization and presentation (4mQY, FZa9, JLdp), and thoroughness of empirical evidence (FZa9, 4mQY, oCzb).

The primary concerns raised by the original round of reviews can be grouped as (1) dependence on privileged information in simulation, (2) fit/relevance to robot learning and CoRL, and (3) disagreements regarding use of the terms “greedy” and “search”. We conducted new experiments, **including a real-world deployment** of LFG and baselines  (without any privileged information), and provide clarifications to various questions as replies to the original reviews.

- [Reviewer FZa9](https://openreview.net/forum?id=PsV65r0itpo&noteId=nw5Ghlr5na) responded that the additional experiments and real-world results “are impressive and a welcome addition to the paper”. Their remaining concern is not with the methodology, claims, or evidence, but with the semantics of usage of terms  “search”, “planning”, and “greedy”. We acknowledge that these terms are overloaded, provide some clarification and references to prior works that have used similar terminology, and commit to incorporating their suggestions for better reception of our research in the broader communities of machine learning and robotics.
- Reviewers [oCzb](https://openreview.net/forum?id=PsV65r0itpo&noteId=weuFqXspKn5) and [JLdp](https://openreview.net/forum?id=PsV65r0itpo&noteId=i0SxgWyptt) said they were satisfied with the rebuttal and had no further concerns.
- [Reviewer 4mQY](https://openreview.net/forum?id=PsV65r0itpo&noteId=gXgz0dI0IJ) said their concerns about unfair baselines were resolved and that “the evaluation was extensive and compelling, and the newly-added experiment in a real-world setting strengthens this claim even further”. They still hold on to their concern about fit. [We argue](https://openreview.net/forum?id=PsV65r0itpo&noteId=lBTLbxgro5) that _”Combination of learning- and planning-based approaches in robotics”_ is explicitly listed in the CoRL call for papers, and CoRL (and similar venues) has set precedence by publishing papers that explore how robotic systems can leverage internet-scale pre-trained models to learn useful behavior without collecting robot-specific data [CLIP-Fields, VL-Maps, Code-as-Policies, ViperGPT, ConceptFusion, CLIP-Nav, DetGPT, NavGPT, Dall-E-Bot, and many more]. None of the other reviewers identify with this concern, and 4mQY also acknowledges that the work is well evaluated, and has a strong robot demonstration in the real-world.


The new real-world experiments and results are described here: https://sites.google.com/view/lfg-anon/home

---

### Decision · Program_Chairs · 2023-08-30

**Decision:**

Accept (Poster)

**Comment:**

This paper concerns navigation in unknown environments and proposes the use of a frozen LLMs to rank exploration goals by the probability that a candidate will lead to the goal.
The core contribution lies in how to appropriately poll the LLM (CoT, +ve/-ve queries etc.) and propose a ranking function (Eq. 1) that includes LLM outputs.

During the rebuttal phase the authors provided clarifications on the baselines and included a real robot experiments. If accepted, authors are requested to carefully incorporate the suggestions provided by the reviewers.

Further, concerns emerged with regard to whether the work is suitable for CoRL since direct use of frozen LLMs departs from prevailing practice of explicitly training models and deploying them on robots.  Since the understanding, methodology and use of LLMs is still emerging it may be prudent to  engage inclusively in the interim. This question merits a broader discussion in the wider CoRL community.